# Three-Dimensional Visualization Using Proportional Photon Estimation Under Photon-Starved Conditions

**DOI:** 10.3390/s25030893

**Published:** 2025-02-01

**Authors:** Jin-Ung Ha, Hyun-Woo Kim, Myungjin Cho, Min-Chul Lee

**Affiliations:** 1Graduate School of Computer Science and Systems Engineering, Kyushu Institute of Technology, 680-4 Kawazu, Iizuka-shi 820-8502, Japan; ha.jinung663@mail.kyutech.jp (J.-U.H.); kim.hyunwoo547@mail.kyutech.jp (H.-W.K.); 2School of ICT, Robotics, and Mechanical Engineering, Hankyong National University, IITC, 327 Chungang-ro, Anseong 17579, Republic of Korea

**Keywords:** digital image processing, integral imaging, photon-counting integral imaging, synthetic aperture integral imaging, volumetric computational reconstruction

## Abstract

In this paper, we propose a new method for three-dimensional (3D) visualization that proportionally estimates the number of photons in the background and the object under photon-starved conditions. Photon-counting integral imaging is one of the techniques for 3D image visualization under photon-starved conditions. However, conventional photon-counting integral imaging has the problem that a random noise is generated in the background of the image by estimating the same number of photons in entire areas of images. On the other hand, our proposed method reduces the random noise by estimating the proportional number of photons in the background and the object. In addition, the spatial overlaps have been applied to the space where photons overlap to obtain the enhanced 3D images. To demonstrate the feasibility of our proposed method, we conducted optical experiments and calculated the performance metrics such as normalized cross-correlation, peak signal-to-noise ratio (PSNR), and structural similarity index measure (SSIM). For SSIM of 3D visualization results by our proposed method and conventional method, our proposed method achieves about 3.42 times higher SSIM than conventional method. Therefore, our proposed method can obtain better 3D visualization of objects than conventional photon-counting integral imaging methods under photon-starved conditions.

## 1. Introduction

Three-dimensional (3D) image visualization techniques have been studied in a variety of fields to enhance the sensing and visualization of real objects. The field of 3D image visualization is broad, and applications include autonomous driving, augmented reality, astrophotography, biomedical, and so on [1,2,3,4,5,6,7,8,9,10,11,12,13,14,15,16,17,18,19]. Moreover, 3D image visualization techniques have been developed, LIDAR sensors [4] have been applied to autonomous driving for measuring the distance, and photon-counting computed tomography (PCCT) [5] has been used to estimate the individual incident X-ray photons and measure their energy levels.

Recently, researchers have studied the reconstruction of 3D integral imaging using incoherent light with a normal camera. Integral imaging is one of the several approaches used to implement 3D visualization techniques [6,7,8,9,10,11,12,13,14]. In 1908, G. Lippmann [6] was awarded the Nobel Prize in Physics for first proposing integral imaging, a technique for 3D visualization from multi-perspective images. It uses two-dimensional (2D) images from different perspectives captured by a lenslet array or camera array, which are referred to as the elemental images. Integral imaging can provide the full parallax and continuous viewpoints without the need for the special observation devices and the coherent light source. However, integral imaging methods based on lenslet array have the problem that the resolution of the elemental image is limited due to the division by the number of lenslets, and visualization of 3D images in low-resolution with the shallow depth of focus [11].

To solve the low resolution and the shallow depth of focus in integral imaging with a lenslet array, synthetic aperture integral imaging (SAII) was proposed [9,10,11]. SAII is one of the camera array-based integral imaging methods. It can record the elemental images with the same resolution as the image sensor, allowing SAII to obtain the high-resolution elemental images. In addition, 3D images can be reconstructed by applying volumetric computational reconstruction (VCR) [12,13,14,18] with the high-resolution elemental images obtained by SAII. It can obtain 3D information by shifting and overlapping the elemental images at the reconstruction plane. However, there is a problem that 2D elemental image of SAII may not be visualized under photon-starved conditions because photons reflected from the object may not reach the image sensor. Therefore, the resolution of the 3D reconstructed image with VCR may be degraded.

Photon-counting integral imaging has been proposed to visualize 3D objects under photon-starved conditions [15,16,17,18,19,20,21,22,23,24]. It is a 3D image visualization technique based on Poisson distribution under photon-starved conditions, calculating the probabilistic statistics of photons that rarely occur in the unit time and space [17]. However, photon-counting integral imaging applies a Poisson random process by estimating the same number of photons in all image areas. As a result, this method has the problem of generating random noise from both the object and the background [18,24].

To solve this problem, we propose a new photon-counting integral imaging method that visualizes objects by proportionally estimating the number of photons reflected from an object under photon-starved conditions in this paper.

In our proposed method, we can proportionally estimate the number of photons based on the generated intensity map by calculating the overall intensity of the elemental image. As a result, we can reduce the random noise by estimating the photons only in the object areas. Furthermore, we can improve the quality of the elemental image by applying the spatial overlaps to the space where the photons overlap to accurately estimate the number of photons in the object. Finally, VCR can be applied to reconstruct a high-resolution 3D image.

This paper is organized as follows. In Section 2, we describe a theory of the integral imaging method, photon-counting integral imaging. Our proposed method, proportional photon-counting integral imaging, is described in Section 3. Then we present the optical experiments and show the results in Section 4. Finally, we present conclusions in Section 5.

## 2. Theory

### 2.1. Integral Imaging

In 1908, Lippmann reported on the passive 3D visualization technique of integral imaging [6]. Figure 1 illustrates the basic concept of integral imaging by the lenslet array.

As shown in Figure 1, integral imaging can be divided into the pickup process and the reconstruction process. The pickup process can take the multiple 2D images with different perspectives using the lenslet array. These 2D images are referred to as the elemental images. The reconstruction process can visualize a 3D image based on the elemental images. Reconstruction methods can be divided into the optical reconstruction and the computational reconstruction. The optical reconstruction can reconstruct a 3D image by projecting the elemental images through the homogeneous lenslet array used in the pickup process [18]. The computational reconstruction can reconstruct 3D image by projecting elemental image through the virtual pinhole array onto the reconstruction plane, which is referred to as volumetric computational reconstruction (VCR) [12,13,14,18].

The integral imaging method based on the lenslet array, as shown in Figure 1, reduces the resolution of the elemental image due to the division by the number of lenslets. This problem can be solved by integral imaging by the camera array. Figure 2 describes the concept of integral imaging by the camera array. Integral imaging based on the camera array can obtain the high-resolution reconstruction image because it can record the elemental images with the same resolution as the image sensor. However, it is difficult to utilize the multiple camera arrays in uniform alignment, and it is not cost-effective. To solve this problem, synthetic aperture integral imaging (SAII) has been proposed [9,10,11]. In SAII, a single camera moves equally spaced horizontally and vertically during the pickup process to acquire the high-resolution elemental images with different perspectives. As a result, a single camera can be used to achieve the same effect as using a camera array, and it is cost-effective. In this paper, we use SAII in the pickup process.

Volumetric computational reconstruction (VCR) can be used to obtain 3D images. Figure 3 shows the basic concept of VCR. It back-projects the elemental images through the virtual pinhole array. In the reconstruction plane, the back-projected elemental images are enlarged and overlap each other. As a result, we can obtain 3D images at different reconstruction depths. The shifting pixels are varied at various reconstruction depths. The shifting pixel value is defined as follows [12,13,14,18]:(1)$$\Delta S_{x}=\frac{N_{x}\times p_{x}\times f}{C_{x}\times Z_{d}},\;\Delta S_{y}=\frac{N_{y}\times p_{y}\times f}{C_{y}\times Z_{d}}$$
where $\Delta S_{x}$ and $\Delta S_{y}$ are the number of the shifting pixels in the *x* and *y* directions on the reconstruction plane. $C_{x}$ and $C_{y}$ are the sizes of the image sensor. $Z_{d}$ is the reconstruction depth. $N_{x}$ and $N_{y}$ are the number of pixels in the elemental image. $p_{x}$ and $p_{y}$ are the pitch between virtual pinholes, and *f* is the focal length of the lenslet array or the distance between the elemental image and the virtual pinhole array [18]. To utilize the shifting pixel values calculated by Equation (Equation 1) in VCR, these values are approximated as the pixel unit by follows [14]:(2)$$\Delta S_{x}^{k}=\left\lceil k\times \Delta S_{x}\right\rfloor ,\;for\;k=0,1,2,\ldots ,K-1$$(3)$$\Delta S_{y}^{l}=\left\lceil l\times \Delta S_{y}\right\rfloor ,\;for\;l=0,1,2,\ldots ,L-1$$(4)$$I_{VCR}\left(x,y,Z_{d}\right)=\frac{1}{O_{VCR}\left(x,y,Z_{d}\right)}\sum_{k=0}^{K-1}\sum_{l=0}^{L-1}E_{kl}\left(x+\Delta S_{x}^{k},y+\Delta S_{y}^{k}\right)$$
where $\Delta S_{x}^{k}$ and $\Delta S_{y}^{l}$ are the rounded shifting pixels of the elemental images in the *k*th column and *l*th row, and *K* and *L* are the number of elemental images in the *x* and *y* directions. $O_{VCR}\left(x,y,Z_{d}\right)$ is the overlapping matrix based on the reconstruction depth. $E_{kl}$ is the elemental image in the *k*th column and *l*th row. $I_{VCR}\left(x,y,Z_{d}\right)$ is a 3D reconstruction image by the nonuniform VCR. Thus, we can obtain a 3D reconstructed image. However, 3D visualization may not be possible under photon-starved conditions because photons may not reach the image sensor, which degrades the intensity of the elemental image. Therefore, photon-counting integral imaging has been proposed to reconstruct 3D images even under photon-starved conditions.

### 2.2. Photon-Counting Integral Imaging

Image sensors can detect the photons reflected from the object based on the photoelectric effect. Therefore, when photons cannot reach the image sensor, the object cannot be visualized. Photon-counting integral imaging techniques can be one of the alternatives to visualizing 3D images under photon-starved conditions [15,16,17,18,19]. Photon-counting integral imaging estimates the photons by applying the probabilistic method to the elemental image. Furthermore, the probabilities of the estimated photons can be improved by applying the maximum likelihood estimation (MLE) [17,18,20,21] and Bayesian approaches such as the maximum a posteriori (MAP) [19,22,23,24]. Thus, we can visualize the elemental image by estimating probabilistically improved photons and reconstructing a 3D image through the computational reconstruction algorithms. Photon-counting integral imaging can be defined as follows [15,16,17,18,19,24]:(5)$$\lambda_{n}\left(x,y\right)=\frac{N_{p}I_{n}\left(x,y\right)}{\sum_{x=1}^{N_{x}}\sum_{y=1}^{N_{y}}I_{n}\left(x,y\right)}$$(6)$$C_{n}\left(x,y\right)|\lambda_{n}\left(x,y\right)∼Poisson\left[\lambda_{n}\left(x,y\right)\right]$$
where $I_{n}\left(x,y\right)$ is a 2D elemental image, $N_{p}$ is the number of photons estimated in the elemental image, and $\lambda_{n}\left(x,y\right)$ is the normalized intensity of the single elemental image, respectively. In addition, $\lambda_{kl}$ is the normalized intensity of multiple elemental images acquired during the pickup process, where *k* and *l* are the indices of the elemental image. Thus, $\lambda_{kl}$ is the normalized intensity of the elemental image in the *k*th column and *l*th row. This means that the normalized intensity of $\lambda_{n}\left(x,y\right)$ has the unit energy by Equation (Equation 5). In addition, under photon-starved conditions, we can assume that photons follow Poisson distribution because they rarely occur in unit time and space [17]. $C_{n}\left(x,y\right)$ is the photon-counting image, which is the estimate of the photons from the Poisson distribution. We applied MLE to determine the occurrence of photons, and the likelihood function and log-likelihood function of MLE are defined as follows [17,18,20,21]:(7)$$P\left(C_{kl}|\lambda_{kl}\right)=\prod_{k=0}^{K-1}\prod_{l=0}^{L-1}\frac{e^{-\lambda_{kl}}{\left(\lambda_{kl}\right)}^{C_{kl}}}{C_{kl}!},$$(8)$$L\left(\lambda_{kl}|C_{kl}\right)\propto \sum_{k=0}^{K-1}\sum_{l=0}^{L-1}\left\{C_{kl}\log\left[\lambda_{kl}\right]\right\}-\sum_{k=0}^{K-1}\sum_{l=0}^{L-1}\lambda_{kl},$$
where $P\left(C_{kl}|\lambda_{kl}\right)$ is the likelihood function of the probability that the photon-counting image arises from the normalized intensity of the elemental image, and $L\left(\lambda_{kl}|C_{kl}\right)$ is the log-likelihood function, which is the log-transformation of the likelihood function, respectively. Using the MLE to maximize the log-likelihood function can be defined as follows [17,18,20,21]:(9)$$\frac{\partial L\left(\lambda_{kl}|C_{kl}\right)}{\partial \lambda_{kl}}=\frac{C_{kl}}{\lambda_{kl}}-N_{p}=0,$$(10)$$\lambda_{kl}^{MLE}=\frac{C_{kl}}{N_{p}},$$

In Equations (Equation 9) and (Equation 10), we can define $\lambda_{kl}^{MLE}$, where $\lambda_{kl}$ is maximized by the partial derivative of $L\left(\lambda_{kl}|C_{kl}\right)$ by $\lambda_{kl}$. As a result, we can estimate photons by applying the MLE under photon-starved conditions as shown in Figure 4.

Figure 4 illustrates the application of MLE to photon-counting imaging. However, MLE can be inaccurate in estimating photons because it uses a uniform distribution as a prior probability with an arbitrary estimate of the number of photons. This means that the probability of the photon occurring has the same probability in all areas. Therefore, specific prior probabilities are required to accurately estimate photons. Based on Bayes’ theorem, the MAP method combines the likelihood function and the prior probability to maximize the posterior probability. In this way, the MAP method can accurately estimate the probability of a photon occurring based on specific prior information [19,22,23,24].(11)$$\pi \left(N_{p}\lambda_{kl}\right)=\frac{\beta^{\alpha }}{\Gamma \left(\alpha \right)}{\left(N_{p}\lambda_{kl}\right)}^{\alpha -1}e^{-\beta N_{p}\lambda_{kl}},\;N_{p}\lambda_{kl}>0$$(12)$$\mu =\frac{\alpha }{\beta },\;\sigma^{2}=\frac{\alpha }{\beta^{2}}\;\Rightarrow \;\alpha =\frac{\mu^{2}}{\sigma^{2}},\;\beta =\frac{\mu }{\sigma^{2}}$$(13)$$\pi \left(N_{p}\lambda_{kl}|C_{kl}\right)∼Gamma\left(C_{kl}+\alpha ,\left(1+\beta \right)\right),$$
where $\pi \left(N_{p}\lambda_{kl}\right)$ is the prior probability distribution of the normalized elemental image with an arbitrary number of photons, which is the conjugate prior distribution of the Poisson distribution. α and β are parameters of the prior probability in the Gamma (Γ) distribution, and both parameters are positive. μ and $\sigma^{2}$ are the mean and variance. In Equation (Equation 13), $\pi \left(N_{p}\lambda_{kl}|C_{kl}\right)$ is modeled as a conjugate distribution to easily calculate the Γ distribution [19,22,23,24].(14)$$\lambda_{kl}^{MAP}=\frac{C_{kl}+\alpha }{N_{p}\left(1+\beta \right)},\;C_{kl}>0$$
where $\lambda_{kl}^{MAP}$ is the estimated intensity of the normalized elemental image using the MAP. By using Equations (Equation 11)–(Equation 14), we can define the MAP that maximizes the posterior probability by combining the likelihood function and the prior probability. As a result, by applying the MAP, we can more accurately estimate the number of photons under photon-starved conditions.

Figure 5 shows the estimated images by applying MLE and MAP under photon-starved conditions. For the same number of photons, the MAP can estimate the image more accurately than the MLE.

Figure 6 shows the noise-estimated image by applying the MAP based on an arbitrary number of photons. Histogram equalization was used to visualize the photon random noise. As shown in Figure 6, the MAP method can estimate photons more accurately than MLE under photon-starved conditions. However, the MAP method generates the random noise because it applies Poisson distribution to all areas by estimating the same number of photons. As a result, the random noise occurs randomly in the background and objects, which degrades the quality of the image. To solve this problem, we propose a method for 3D visualization by proportionally estimating the number of photons in this paper.

## 3. Three-Dimensional Reconstruction by Proportional Estimation of Photon-Counting Method Under Photon-Starved Conditions

### 3.1. Proportional Photon-Counting Imaging (PPCI)

The conventional photon-counting imaging method has the problem of the random noise occurring by applying the Poisson distribution to entire areas. To solve this problem, we propose the proportional photon-counting imaging (PPCI) method, which estimates the number of photons proportionally rather than estimating the number of photons in entire areas. To proportionally estimate the number of photons, we calculate the overall intensity of the elemental image and the overall intensity of the kernel to create the proportional intensity map. The proportional intensity map calculates proportionally the intensity for the objects in the elemental image. In addition, by normalizing the generated intensity map, we can proportionally estimate the number of photons and estimate the location where the object exists. As a result, we can reduce the random noise by proportionally estimating the number of photons in the background and the object. In addition, the spatial overlapping can be used to improve the quality of the elemental image in the space where photons overlap. The spatial overlapping divides the elemental image into multiple non-overlapping regions the size of the kernel, and the accurate photon counting can be estimated by the counted overlapping photons as the kernel is shifted. The method of proportionally estimating the number of photons can be defined as follows:(15)$$E_{inten}^{kl}=\sum_{n_{x}=0}^{N_{x}-1}\sum_{n_{y}=0}^{N_{y}-1}E_{kl}\left(n_{x},n_{y}\right)$$(16)$$K_{inten}^{wh}=\sum_{ks_{x}=0}^{KS_{x}-1}\sum_{ks_{y}=0}^{KS_{y}-1}K_{wh}\left(ks_{x},ks_{y}\right)$$(17)$$\psi_{map}\left(x,y\right)=\sum_{w=0}^{W-1}\sum_{h=0}^{H-1}\frac{K_{inten}^{wh}}{E_{inten}^{kl}}\times 100$$(18)$$E_{p}=\frac{\psi_{map}\left(x,y\right)-\min\left(\psi_{map}\left(x,y\right)\right)}{\max\left(\psi_{map}\left(x,y\right)\right)-\min\left(\psi_{map}\left(x,y\right)\right)}\times N_{p}$$
where $E_{kl}\left(n_{x},n_{y}\right)$ is the elemental image, and $E_{inten}^{kl}$ is the overall intensity of the elemental image. In other words, Equation (Equation 15) calculates the overall intensity of the elemental images with the size of $N_{x}$ and $\;N_{y}$ for the elemental images existing in the *k*th column and *l*th row position. $K_{wh}\left(ks_{x},ks_{y}\right)$ is the kernel image in the *w*th column and *h*th row of the elemental images divided into non-overlapping regions the size of the kernel, and $K_{inten}^{kl}\left(x_{k},y_{k}\right)$ is the overall intensity of $K_{wh}\left(ks_{x},ks_{y}\right)$. $KS_{x}$ is the size of the kernel in the horizontal direction, and $KS_{y}$ is the size of the kernel in the vertical direction. In other words, Equation (Equation 16) calculates the overall intensity for kernels of a specific size that exist in the *w*th column and *h*th row position. $\psi_{map}\left(x,y\right)$ is the proportional intensity map, which is the proportional calculation of the intensity of the elemental image. $E_{p}$ is the number of photons estimated proportionally by normalizing $\psi_{map}\left(x,y\right)$. $N_{p}$ is the number of photons estimated in the elemental image, respectively. As defined in Equations (Equation 15)–(Equation 18), we can generate the proportional intensity map, based on which we can proportionally estimate the number of photons. Thus, we can accurately estimate the number of photons reflected from the object. Using the proportionally estimated number of photons, we apply photon-counting imaging with MAP to visualize the image.

Figure 7 shows the proportional intensity map generated by proportionally calculating the intensity of the elemental image for each color channel. Finally, we can obtain the better estimated image by our proposed method as follows:(19)$${\hat{\lambda }}_{wh}^{MAP}=\frac{{\hat{C}}_{wh}+\alpha }{E_{p}\left(1+\beta \right)},\;E_{p}\geq 0$$(20)$${\tilde{C}}_{wh}|{\hat{\lambda }}_{wh}^{MAP}∼Poisson\left({\hat{\lambda }}_{wh}^{MAP}\right)$$(21)$$\Delta KSP_{x}^{w}=\left\lceil w\times KSP_{x}\right\rfloor ,\;for\;w=0,1,2,\ldots ,W-1$$(22)$$\Delta KSP_{y}^{h}=\left\lceil h\times KSP_{y}\right\rfloor ,\;for\;h=0,1,2,\ldots ,H-1$$(23)$$O_{spatial}\left(x,y\right)=\sum_{w=0}^{W-1}\sum_{h=0}^{H-1}1\left\{x+\Delta KSP_{x}^{w},\;y+\Delta KSP_{y}^{h}\right\}$$(24)$$R\left(x,y\right)=\frac{1}{O_{spatial}\left(x,y\right)}\sum_{w=0}^{W-1}\sum_{h=0}^{H-1}{\tilde{C}}_{wh}\left\{x+\Delta KSP_{x}^{w},\;y+\Delta KSP_{y}^{h}\right\}$$
where ${\hat{\lambda }}_{wh}^{MAP}$ is the estimated intensity of the normalized elemental image by applying MAP based on the proportionally estimated number of photons. ${\tilde{C}}_{wh}$ is the proportional estimation photon-counting image by applying ${\hat{\lambda }}_{wh}^{MAP}$. $\Delta KSP_{x}^{w}$ and $\Delta KSP_{y}^{h}$ are the rounded shifting pixels of the kernel moved along the *x* and *y* directions. *W* and *H* are the number of positions for kernel in the *x* and *y* directions. 1 is the ones matrix. $O_{spatial}\left(x,y\right)$ is the spatial overlapping matrix of the space in which photons overlap as the kernel that is shifted in the elemental image divided into regions. $R\left(x,y\right)$ is the reconstructed elemental image, where photon-counting is applied based on the proportionally estimated number of photons, with spatial overlaps applied to the space where photons overlap [24].

Figure 8 shows the reconstructed result of PPCI. We can generate the intensity map by proportionally calculating the overall intensity of the elemental image, based on which we can estimate the number of photons. Furthermore, we can visualize the image using photon-counting MAP, applying spatial overlapping to the space where the photons overlap. Finally, we can visualize a 3D image by applying VCR.

### 3.2. Proportional Estimation Integral Imaging (PEII)

In this section, we propose a 3D reconstruction method using the VCR method based on the elemental image visualized by applying the PPCI method. We refer to this integral imaging method as Proportional Estimation Integral Imaging (PEII). Figure 9 shows an overview of the PEII method.

As shown in Figure 9, under photon-starved conditions, the elemental images are obtained by performing the pickup process with the SAII method. The acquired elemental image cannot visualize the image due to the insufficient number of the estimated photons. We apply PPCI to the elemental images that cannot be visualized. As a result, we can reduce the noise in the elemental images by proportionally estimating the number of photons, and we can visualize the image by improving the quality of the elemental images. Finally, we can reconstruct the 3D image with the VCR method. The PEII method can be defined by the following equation.(25)$${\tilde{I}}_{PEII}\left(x,y,Z_{d}\right)=\frac{1}{\tilde{O}\left(x,y,Z_{d}\right)}\sum_{w=0}^{W-1}\sum_{h=0}^{H-1}{\tilde{R}}_{wh}\left(x+\Delta S_{x}^{w},y+\Delta S_{y}^{h}\right)$$
where ${\tilde{R}}_{wh}$ is the elemental image reconstructed by applying PPCI, which is the elemental image in the *w*th column and *h*th row. ${\tilde{I}}_{PEII}\left(x,y,Z_{d}\right)$ is the reconstructed 3D image by PEII with the nonuniform VCR.

Figure 10 shows the enlarged part of the reconstructed results by applying PEII. To demonstrate the feasibility of our proposed methods, we describe the optical experiments and results in the next sections.

## 4. Experimental Setup and Results

In this section, we describe the experimental setup for applying proportional estimation integral imaging and photon-counting integral imaging methods to elemental images acquired through SAII. In addition, we show the experimental results and demonstrate the feasibility of our proposed method.

### 4.1. Experimental Setup

In this experimental setup, we use SAII to obtain elemental images. SAII can reproduce the 2D camera array by moving the single camera (D850, Nikon, Tokyo, Japan) in horizontal and vertical directions to record elemental images. This single camera captures the elemental image by moving on the position of the camera array described in Figure 11a and in Section 2.1. The experimental setups of our proposed method can be applied under two conditions. The first is the environmentally limited light condition. This is the condition in which the intensity of light is limited; the reflecting photons from the object reaching the camera are limited. The second is the mechanically limited light condition of the camera. This is the condition in which the number of photons reaching the image sensor is limited by controlling the shutter speed of the camera. In this experimental setup, we proceeded under the second condition. Figure 11 illustrates the experimental setup and the elemental images obtained by the experimental setup. In Figure 11a, we use a 5×5 camera array to take elemental images, and the pitch between cameras is 2 mm. Thus, the total number of elemental images is 25, and the resolution of each elemental image is 4128×2752. The objects are positioned at different distances from the camera array. Object 1 is located at 450 mm, object 2 is located at 530 mm, and object 3 is located at 610 mm. Figure 11b shows the elemental image. Table 1 shows the specifications of the experimental setup.

### 4.2. Result

#### 4.2.1. Result of Reconstructing the 2D Elemental Image

Figure 12 shows the results of the conventional method and our proposed method for 2D elemental images under photon-starved conditions. In this subsection, the conventional method is the computational photon-counting imaging. Figure 12a shows a 2D elemental image obtained under normal illumination conditions, and Figure 12b shows a 2D elemental image obtained under photon-starved conditions, and objects cannot be recognized by the human eye. Figure 12c shows a 2D elemental image visualized by applying the conventional methods under the same conditions as shown in Figure 12b. Figure 12d is the elemental image visualized by applying our proposed method. The number of arbitrarily estimated photons in Figure 12c,d is 55% of the total number of pixels (the number of photons is around 6 M). However, the random noise is generated in all areas of the image, and the image cannot be visualized when the number of photons is insufficient in Figure 12c. On the other hand, in Figure 12d, the number of photons can be proportionally estimated to remove the random noise in the background, and the object can be visualized more brightly and accurately.

Figure 13 shows the random noise of the background and objects in the result image visualized by applying the conventional method and our proposed method. Histogram equalization was used to visualize the random noise in the background. Figure 13a shows the result of applying the conventional method. It shows that the conventional method applies the Poisson distribution to all areas, which generates random noise in the background. In addition, the insufficient number of arbitrarily estimated photons cannot accurately recognize the object. Figure 13b shows the visualized result of applying our proposed method. It reduces the random noise in the background by proportionally estimating the number of photons based on the intensity map. Furthermore, the object can be visualized brighter and more accurately by applying spatial overlapping. For a more accurate comparison, we performed a numerical analysis using the performance metrics of normalized cross-correlation, peak signal-to-ratio (PSNR), and structural similarity index measure (SSIM) for each object [25,26,27].

Figure 14 shows the results of the numerical analysis. They are measured by increasing the number of photons in seven steps of 10, 25, 40, 55, 70, 85, and 100% of the total number of pixels. Figure 14a–c show the results of the normalized cross-correlation, PSNR, and SSIM for object 1 (gray vehicle) at 450 mm. In Figure 14c, the average SSIM of our proposed method is 0.171, and the average SSIM of the conventional method is 0.055, which is a difference of approximately 3.11 times. Figure 14d–f show the numerical analysis results of object 2 (green truck) at 530 mm. In Figure 14f, the average SSIM of our proposed method is 0.15, and the average SSIM of the conventional method is 0.067, which is a difference of approximately 2.24 times. Figure 14g–i show the numerical analysis results of object 3 (yellow vehicle) at 610 mm. In Figure 14i, the average SSIM of the proposed method is 0.124, and the average SSIM of the conventional method is 0.088, which is a difference of approximately 1.41 times. As shown in Figure 14, our proposed method shows numerically better results than the conventional method. As a result, under photon-starved conditions, our proposed method can visualize 2D elemental images more clearly and accurately than the conventional methods.

As shown in Figure 14a,c, the proposed method has a lower or decreasing part in the numerical analysis than the conventional method. This is because photons are saturated for object1 after the number of photons is 55%. Figure 15 shows the saturation of object 1 at 55% and 100% of the number of photons. When the number of photons is 55%, we can see that certain parts of the object are saturated, and when the number of photons is 100%, we can see that most parts of the object are saturated.

#### 4.2.2. The Result of Reconstructing the 3D Image

In the previous section, we showed the results of conventional and proposed methods for visualizing 2D elemental images under photon-starved conditions. In this subsection, we compare and analyze the results of the 3D reconstructed image by VCR. Figure 16 shows the results of the 3D reconstruction by applying VCR to a 2D elemental image obtained under normal illumination conditions. Figure 16a–c show the reconstructed 3D images at 450 mm, 530 mm, and 610 mm, respectively.

Figure 17 shows the reconstructed 3D images by applying the conventional method and our proposed method under photon-starved conditions. Figure 17a,b show the reconstructed 3D image by the conventional method and our proposed method at 450 mm, respectively. Figure 17c,d show the reconstructed 3D image at 530 mm, and Figure 17e,f show the reconstructed 3D image at 610 mm, respectively. Here, the number of photons is 55% of the total number of pixels. As a result, our proposed method not only reduces the random noise by proportionally estimating the number of photons but also provides the improved 3D reconstruction by visualizing the object more clearly.

Figure 18 shows the numerical analysis results of the normalized cross-correlation, PSNR, and SSIM for the reconstructed 3D images at different distances. The numerical analysis results are measured by increasing the number of photons in seven steps of 10, 25, 40, 55, 70, 85, and 100% of the total number of pixels. Figure 18a–c show the numerical analysis results for object 1 at 450 mm. In Figure 18c, the average SSIM of our proposed method is 0.301, and the average SSIM of the conventional method is 0.088, which is a difference of approximately 3.42 times. Figure 18d–f show the results for object 2 located at 530 mm. In Figure 18f, the average SSIM of our proposed method is 0.324, and the average SSIM of the conventional method is 0.103, which is a difference of approximately 3.14 times. Figure 18g–i show the results for object 3 located at 610mm. In Figure 18i, the average SSIM of the proposed method is 0.157, and the average SSIM of the conventional method is 0.067, which is a difference of approximately 2.34 times. As shown in Figure 18, our proposed method performs better in numerical analysis than the conventional method.

## 5. Conclusions

In this paper, we have proposed the proportional estimation integral imaging (PEII) method to reconstruct 3D images by proportionally estimating the number of photons under photon-starved conditions. The conventional photon-counting integral imaging method for 3D visualization under photon-starved conditions estimates the same number of photons in all areas to visualize the image, resulting in the random noise and degraded image quality. To solve these problems, our proposed method can proportionally estimate the number of photons based on the intensity map created by calculating the overall intensity of the elemental image. In addition, by applying spatial overlapping to the space where the photons overlap, the number of photons for the object can be accurately estimated. As a result, our proposed method can reduce the random noise and reconstruct the high-resolution 3D images by proportionally estimating the number of photons. Furthermore, our proposed method has been verified through numerical analysis using metrics such as the normalized cross-correlation, PSNR, and SSIM. When our proposed method is applied, the average value of the normalized cross-correlation for each object in the 3D visualization result is 1.19 times higher, the average value of PSNR is 1.25 times higher, and the average value of SSIM is 2.97 times higher. Therefore, our proposed method can obtain better 3D visualization results of objects than the conventional photon-counting integral imaging method under photon-starved conditions. However, there are cases where the performance of our proposed method has been lower than the conventional method in numerical analysis. This is because our proposed method reaches photon saturation with fewer photons than the conventional method. We have considered that the reconstructed 3D image can be obtained more accurately when the number of photons is estimated based on the intensity map and photon saturation map. Our proposed method is expected to contribute to the overall development of technologies that utilize photon-counting algorithms to visualize in 3D imaging techniques, such as autonomous driving systems, astrophotography, medical imaging, and augmented reality technologies.

## Figures and Tables

**Figure 1 sensors-25-00893-f001:**
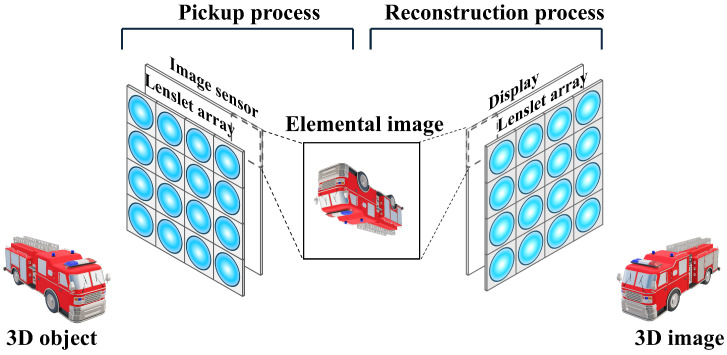
Concept of the integral imaging by the lenslet array.

**Figure 2 sensors-25-00893-f002:**
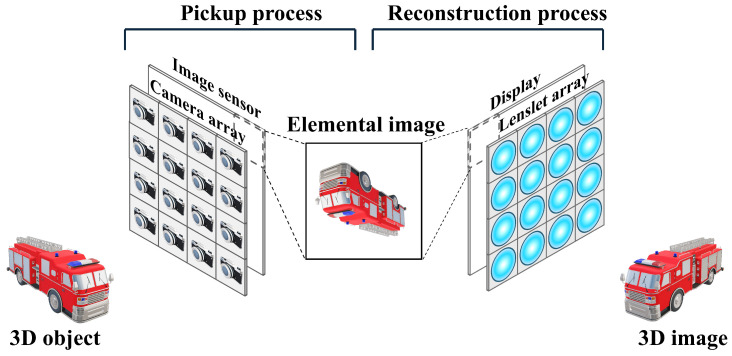
Concept of integral imaging method by the camera array.

**Figure 3 sensors-25-00893-f003:**
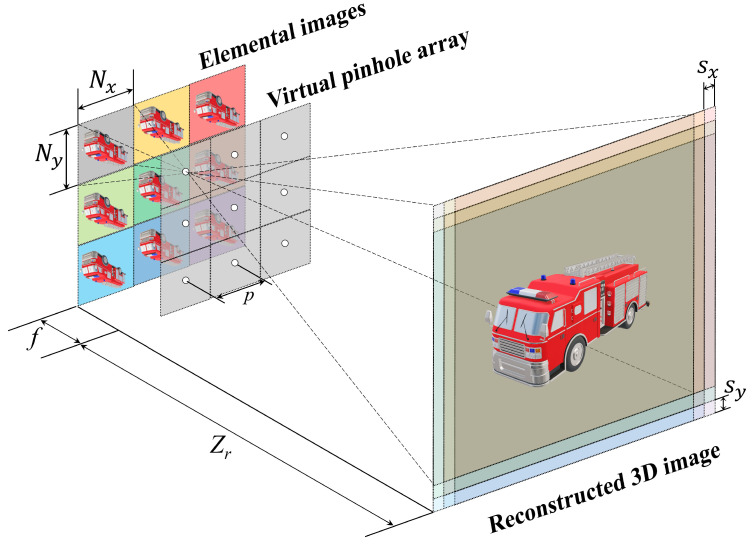
Concept of volumetric computational reconstruction (VCR).

**Figure 4 sensors-25-00893-f004:**
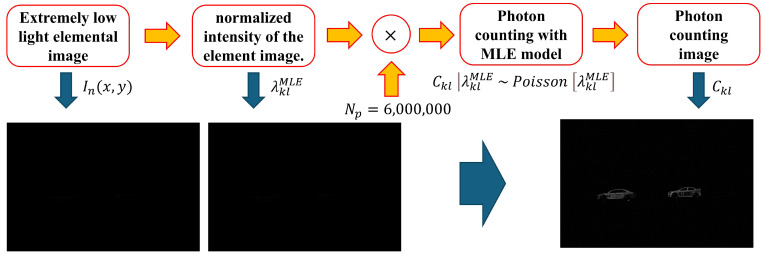
Flowchart of computational photon counting with MLE.

**Figure 5 sensors-25-00893-f005:**
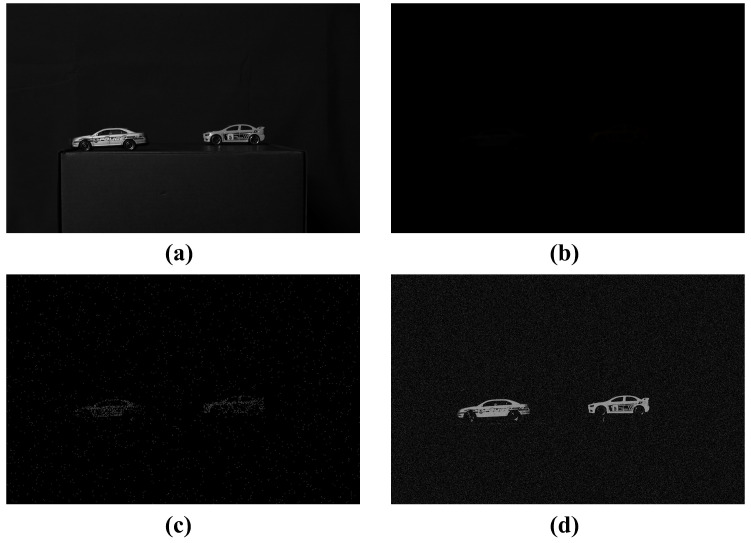
Results of estimated elemental images by applying computational photon counting under photon-starved conditions. (**a**) Reference image, (**b**) image obtained under photon-starved conditions, (**c**) estimated image by photon counting with MLE, (**d**) estimated image by photon counting with MAP, where the number of photons is 3000.

**Figure 6 sensors-25-00893-f006:**
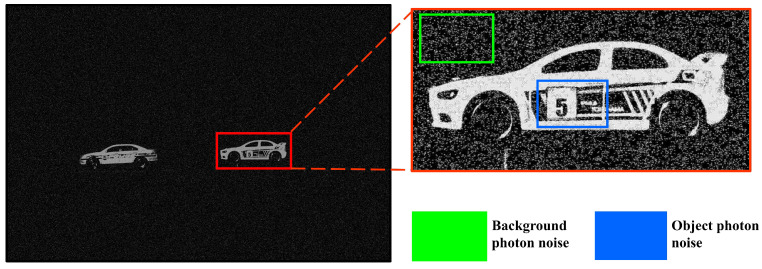
Noise of the estimated image by applying computational photon counting with MAP.

**Figure 7 sensors-25-00893-f007:**
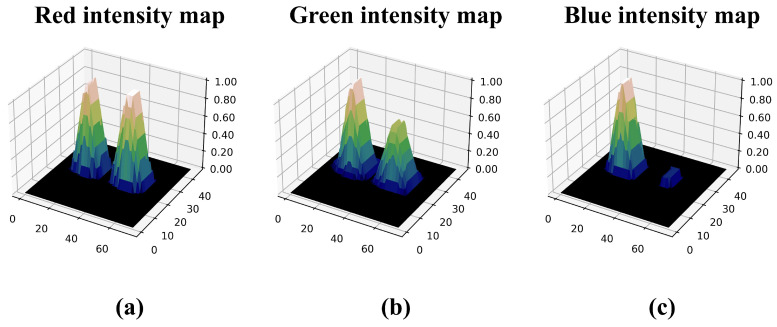
Proportional intensity map for each RGB color channel. (**a**) Intensity map of the red channel, (**b**) intensity map of the green channel, and (**c**) intensity map of the blue channel.

**Figure 8 sensors-25-00893-f008:**
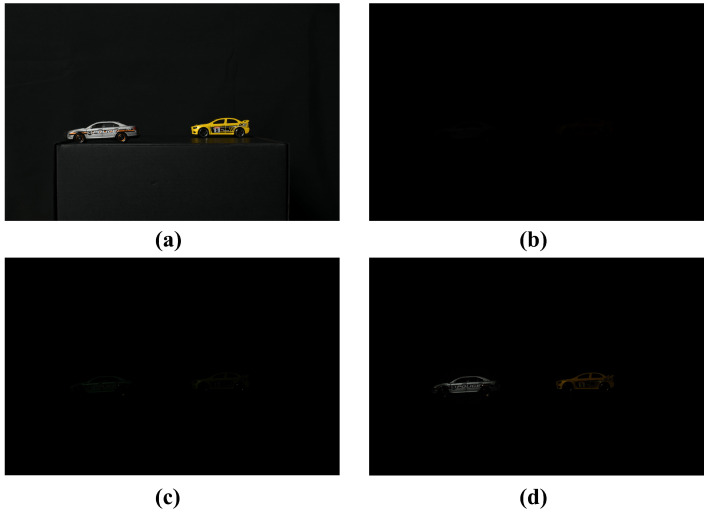
Comparison of results between computational photon counting with MAP and PPCI. (**a**) Reference image, (**b**) image obtained under photon-starved conditions, (**c**) image reconstructed by photon counting with MAP, and (**d**) image reconstructed by PPCI, where the number of photons is 4,000,000.

**Figure 9 sensors-25-00893-f009:**
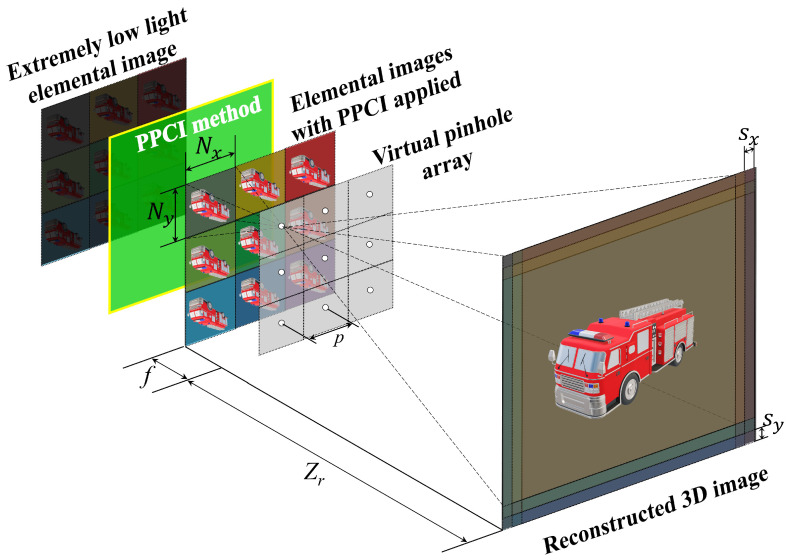
Concept of proportional estimation integral imaging.

**Figure 10 sensors-25-00893-f010:**
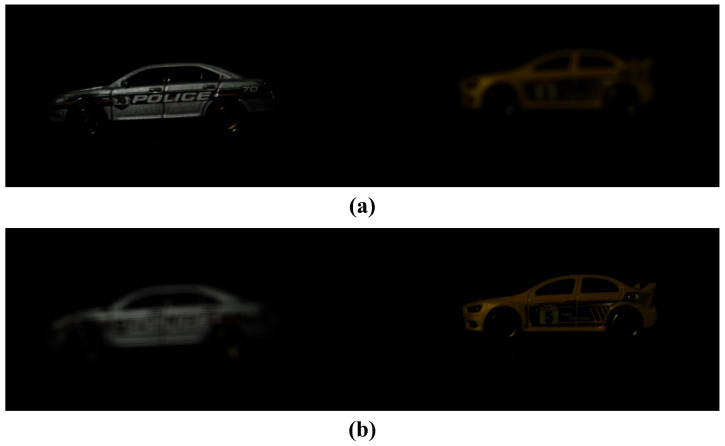
Three-dimensional image reconstructed by applying PEII at (**a**) 450 mm and (**b**) 600 mm.

**Figure 11 sensors-25-00893-f011:**
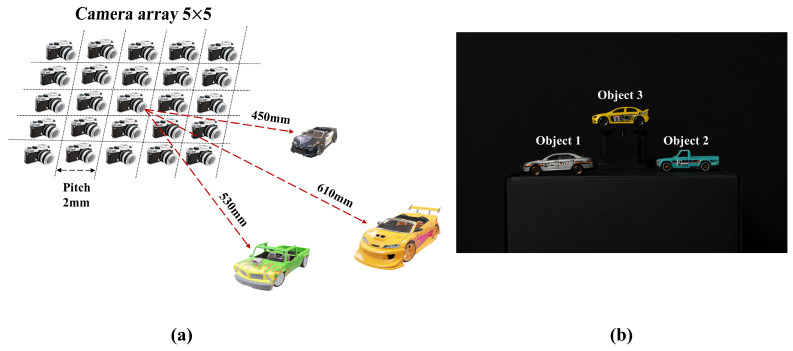
(**a**) Experimental setup and (**b**) elemental image.

**Figure 12 sensors-25-00893-f012:**
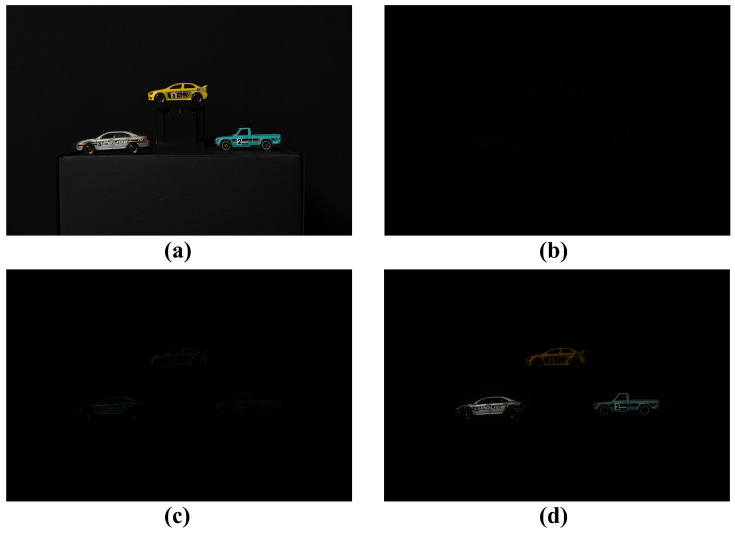
Results of the conventional method and our proposed method for visualizing 2D elemental image. (**a**) Reference image, (**b**) image obtained under photon-starved conditions, (**c**) image reconstructed by conventional method, (**d**) image reconstructed by our proposed method, where the number of photons is 55% of the total number of pixels.

**Figure 13 sensors-25-00893-f013:**
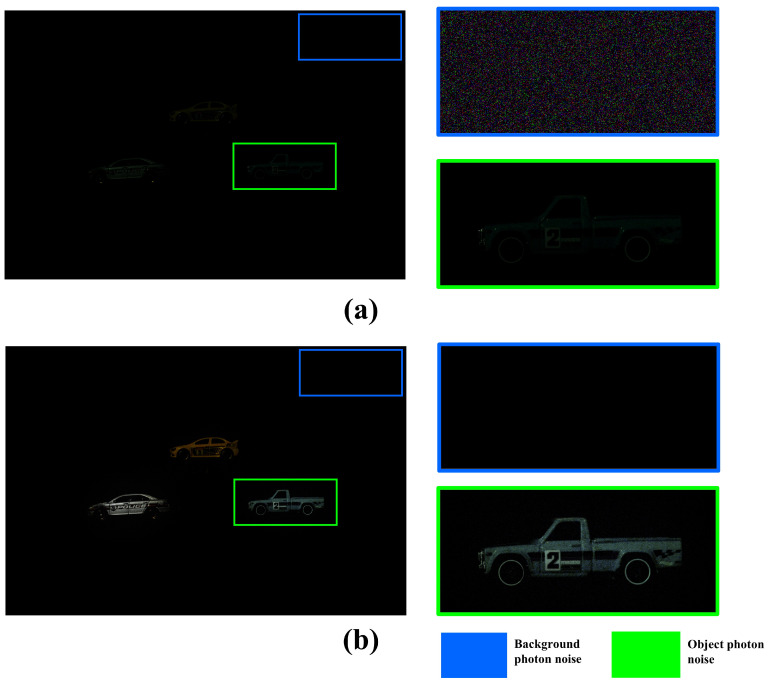
Compare the noise generated by object and background of the visualized 2D elemental image. (**a**) Result of the elemental image by the conventional method and (**b**) result of the elemental image by our proposed method.

**Figure 14 sensors-25-00893-f014:**
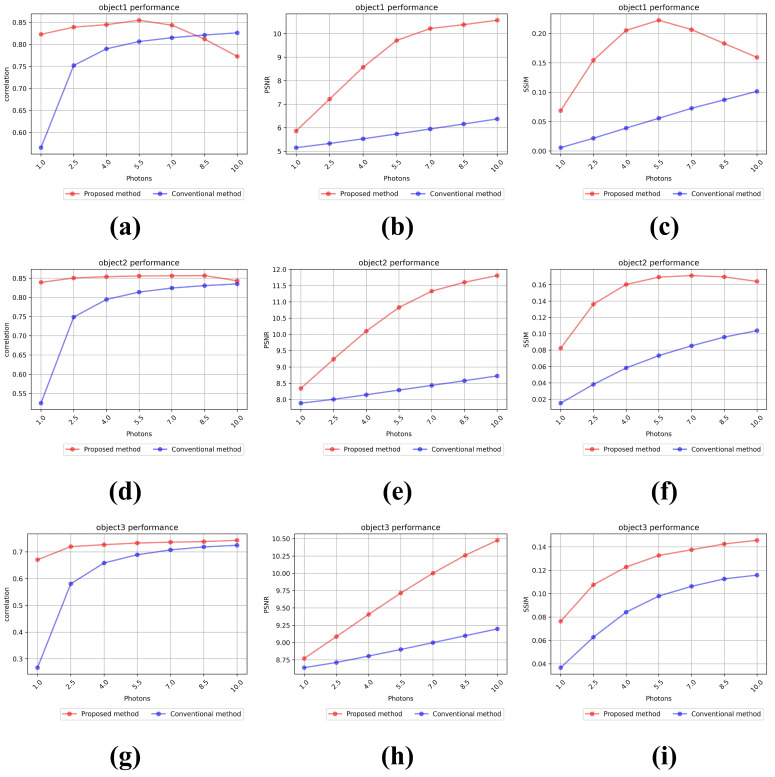
Performance metrics from the numerical analysis for each object. Results of object 1 by (**a**) normalized cross-correlation, (**b**) PSNR, and (**c**) SSIM. Results of object 2 by (**d**) normalized cross-correlation, (**e**) PSNR, and (**f**) SSIM. Results of object 3 by (**g**) normalized cross-correlation, (**h**) PSNR, and (**i**) SSIM.

**Figure 15 sensors-25-00893-f015:**
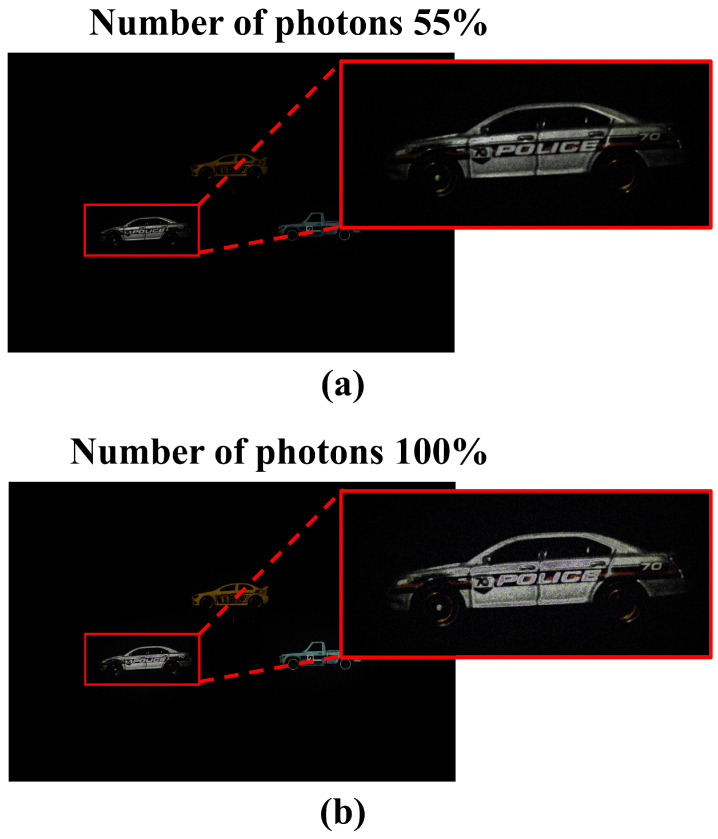
Photon saturation of object 1 by the number of photons. (**a**) Result when the number of photons is 55% and (**b**) result when the number of photons is 100%.

**Figure 16 sensors-25-00893-f016:**
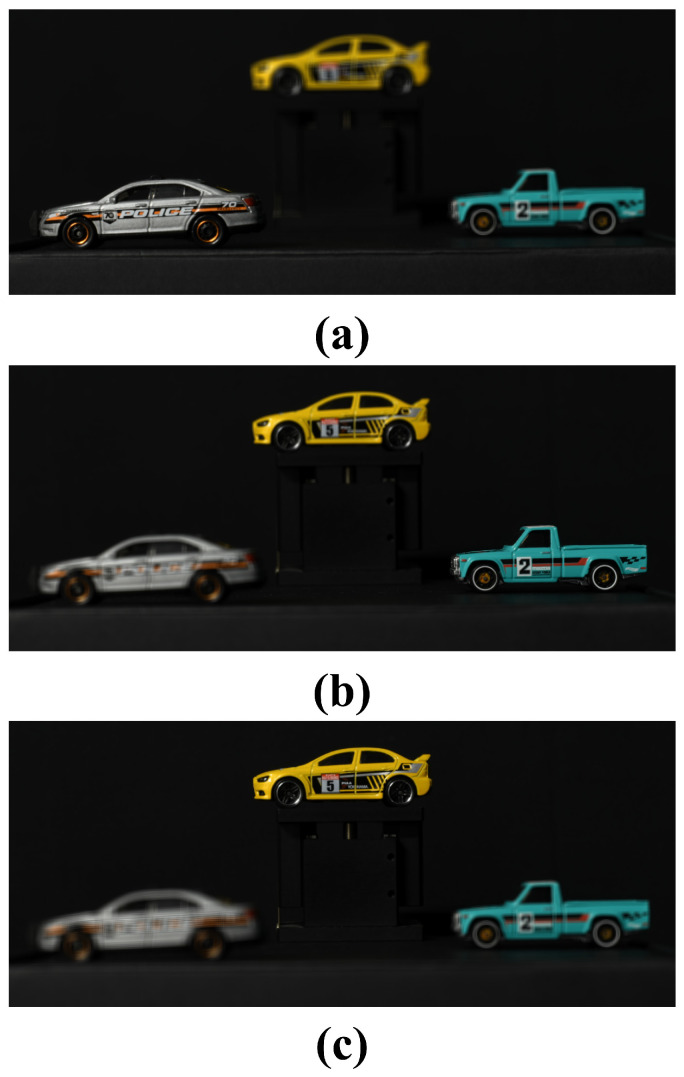
Results by VCR in normal illumination conditions at different distances. (**a**) Reconstructed 3D image at 450 mm, (**b**) reconstructed 3D image at 530 mm, (**c**) reconstructed 3D image at 610 mm.

**Figure 17 sensors-25-00893-f017:**
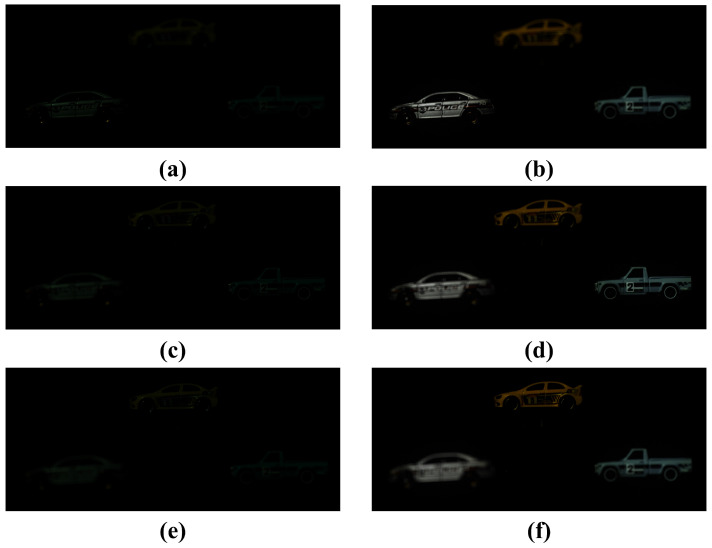
Results of applying the conventional method and our proposed method under photon-starved conditions. (**a**,**c**,**e**) are the result of the reconstructed 3D image using the conventional method at 450 mm, 530 mm, and 610 mm, respectively. (**b**,**d**,**f**) are the result of the reconstructed 3D image using our proposed method at 450 mm, 530 mm, and 610 mm, respectively.

**Figure 18 sensors-25-00893-f018:**
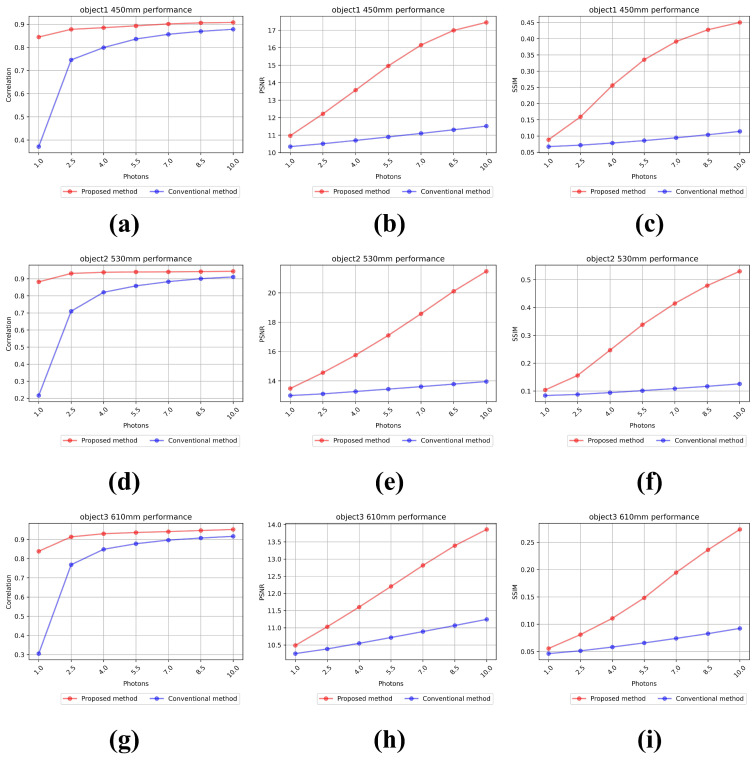
Performance metric for the reconstructed 3D images at each distance. Results of object 1 at 450 mm by (**a**) normalized cross-correlation, (**b**) PSNR, and (**c**) SSIM, Results of object 2 at 530 mm by (**d**) normalized cross-correlation, (**e**) PSNR, and (**f**) SSIM. Results of object 3 at 610 mm by (**g**) normalized cross-correlation, (**h**) PSNR, and (**i**) SSIM.

**Table 1 sensors-25-00893-t001:** Specifications of the experimental setup.

Setup	Nikon D850	
Resolution	4128 × 2752	
Camera array	5 × 5	
Pitch	2 mm	
Sensor size	35.9 × 23.9 mm	
Kernel size	500 × 500	
Focal length	50 mm	
ISO	400	
Shutter speed	Normal	1.6 s
	Extremely low-light	1/5 s

## Data Availability

Data are contained within the article.

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
