# Peer review of "Three-Dimensional Visualization Using Proportional Photon Estimation Under Photon-Starved Conditions"

_sensors, 2025, doi:10.3390/s25030893_

Round 1

Reviewer 1 Report

Comments and Suggestions for Authors

The author addresses the issue of random noise in the background caused by the uniform photon estimation across all regions in traditional photon-counting integral imaging. They propose the Proportional Estimation Integral Imaging (PEII) technique, which reduces random noise by proportionally estimating the number of photons in both the background and target regions. Experimental results, including metrics such as normalized cross-correlation, PSNR, and SSIM, demonstrate the effectiveness of their approach. The paper's structure and logic are relatively complete, but there are several points of confusion or issues as follows:

1.      The author briefly introduces the main content of the paper in the abstract but does not include specific results, such as numerical data. It may be beneficial to add a description of the results and their significance in the abstract.

2.      In section 2.1, the author briefly describes 3D reconstruction methods based on microlens arrays and camera arrays, and mentions that a synthetic aperture method will be used later to obtain the elemental images. However, in section 4.1, the elemental images are obtained using a camera array, which creates a contradiction between the two sections.

3.      In the descriptions of 3D reconstruction in equations 9, 20, and 21, the author does not mention or clearly describe how the spatial overlap matrix O is obtained, which I believe is an important point.

4.      Equations 2 and 3 appear to be missing a space after the word "for."

5.      In section 2.2, the author does not clearly describe the relationships between the parameters in photon-counting integral imaging, nor their specific meanings. For example, what is the relationship between λn and λkl ​? What do the indices k and l specifically represent—are they pixel coordinates or elemental image coordinates? Is λkl ​inconsistent for each pixel in every elemental image? The author should consider providing clearer explanations in this section.

6.      Can it be understood that both the MLE-based and MAP-based methods primarily focus on determining λkl ​, and that the parameter Kkl represents a specific region in the elemental image? The impact of the selection of this value on the final reconstruction result is not discussed.

7.      The author achieves photon starvation conditions by reducing the camera exposure time. However, in applications such as nighttime detection, where the target contrast is inherently low, can this technique still be effective? Additionally, I am curious about how this image processing method differs from traditional image enhancement algorithms.

8.      Lack of quantitative description and prominence of innovation in article abstract and conclusions

Reviewer 2 Report

Comments and Suggestions for Authors

good luck

Round 2

Reviewer 1 Report

Comments and Suggestions for Authors

1. Many images have a large amount of black areas, and the author is expected to make adjustments;

2. No further derivation symbols need to be added before the formula, as in Formula 14;

3. Figure 6 needs to indicate exactly what the noise level is.
